# Guidelines for a Morphometric Analysis of Prokaryotic and Eukaryotic Cells by Scanning Electron Microscopy

**DOI:** 10.3390/cells10123304

**Published:** 2021-11-25

**Authors:** Dominika Czerwińska-Główka, Katarzyna Krukiewicz

**Affiliations:** Department of Physical Chemistry and Technology of Polymers, Silesian University of Technology, 44-100 Gliwice, Poland; dominika.czerwinska-glowka@polsl.pl

**Keywords:** *Escherichia coli*, image analysis, morphological analysis, morphometry, neuroblastoma, scanning electron microscopy

## Abstract

The invention of a scanning electron microscopy (SEM) pushed the imaging methods and allowed for the observation of cell details with a high resolution. Currently, SEM appears as an extremely useful tool to analyse the morphology of biological samples. The aim of this paper is to provide a set of guidelines for using SEM to analyse morphology of prokaryotic and eukaryotic cells, taking as model cases *Escherichia coli* bacteria and B-35 rat neuroblastoma cells. Herein, we discuss the necessity of a careful sample preparation and provide an optimised protocol that allows to observe the details of cell ultrastructure (≥ 50 nm) with a minimum processing effort. Highlighting the versatility of morphometric descriptors, we present the most informative parameters and couple them with molecular processes. In this way, we indicate the wide range of information that can be collected through SEM imaging of biological materials that makes SEM a convenient screening method to detect cell pathology.

## 1. Introduction

A continuous development of biological research is possible due to the constant improvement in experimental techniques. The classical observational and descriptive methods in light microscopy allowed for the visualisation of objects invisible to the naked eye, constituting a breakthrough in microbiology and medicine. The invention of an electron microscopy enabled to observe cell details, including the smallest cell organelles, with a high resolution [1]. Thanks to the continuous advances in scanning electron microscopy (SEM), the observation of the micro world and a deep interpretation of observed images have become achievable.

The first SEM image was taken in 1935 by Max Knoll [2], and now SEM is an extremely useful method in biomedical engineering, allowing a comprehensive observation of isolated microorganisms or cells at a significant magnification [3]. To create a SEM image, an electron beam is emitted from a cathode, accelerated and attracted by a positively charged anode. When accelerated electrons hit the specimen, their kinetic energy is dissipated, which is a source of secondary electrons (typically used to create a SEM image), reflected or back-scattered electrons, characteristic X-rays (used for elemental analysis by means of an energy dispersive spectroscopy), light, heat, and transmitted electrons. The resolution of obtained image depends on electron wavelength, which is much shorter than the wavelength of visible light providing much higher resolution (typically approx. 10 nm) than in traditional light microscopy (about 200–250 nm). New generations of high resolution SEM allow to achieve image resolution of 1 nm, approaching the resolution typical for a transmission electron microscopy, TEM, but without requiring high accelerating potentials (100–200 kV) [4]. Nevertheless, since the best instrumentation is only as good as the best specimen [5], sample preparation should be performed under favourable conditions.

The aim of this paper is to provide a set of guidelines for using a scanning electron microscopy to analyse morphology of prokaryotic and eukaryotic cells, taking as model cases *Escherichia coli* bacteria (known as a molecular biologist tool box [6]) and B-35 rat neuroblastoma cells (an easily transfected, cultured cell model of central nervous system neurons [7]). We discuss the necessity of a careful sample preparation and provide an optimised protocol that allowed to observe the details of ultrastructure (≥ 50 nm) of model cells with a minimum processing effort. Highlighting the versatility of morphometric descriptors that can be used to analyse prokaryotic and eukaryotic cells, we present the most valuable parameters and couple them with molecular processes, such as cell division, mobility, growth dynamics, apoptosis, and necrosis. In this way, we indicate the wide range of information that can be collected through SEM imaging of biological materials.

## 2. Sample Preparation

To obtain a high resolution SEM image of small objects, such as bacteria and eukaryotic cells, proper preparation of a sample is essential. Since it is crucial to maintain a vacuum of at least 10^−4^ Pa and to prevent contamination, biological samples must be carefully processed and dehydrated. An inadequate preparation of a specimen may result in shrinkage and deformation of cells, leading to incorrect results and misleading interpretations [8]. The process of preparing biological samples for SEM imaging consists of several stages, and the main ones include fixation, dehydration and drying. In addition, specimens are typically sputter-coated with a conducting film, since the investigated surface should be conductive to prevent image distortion caused by an electron charging effect [9].

### 2.1. Fixation

In the first stage of sample preparation (fixation), biological samples are treated with protein and/or lipid crosslinking reagents. The aim of this process is to keep the structure of a biological material unchanged, thus enabling the imaging of a natural state of an object. Fixation should also prevent autolysis and decomposition of cells. Many fixatives are known [10], however in practice the most commonly used are glutaraldehyde, osmium tetroxide or their mixtures with other compounds, e.g., paraformaldehyde—the comparison of their properties is presented in Table 1. Generally, the penetration of a fixative into the cell membrane is a long-term process lasting from several to several dozen hours, requiring the use of an appropriately selected pH, temperature, and osmolarity. The basic fixative with the strongest protein crosslinking character is undoubtedly glutaraldehyde. Its effectiveness is related to its multicomponent nature, where at a given pH several forms are present in a reagent solution [11,12,13]. Osmium tetroxide is the second most commonly used biological fixative that crosslinks lipids. However, it penetrates into the cell structure slower than glutaraldehyde. Moreover, its vapours are harmful to eyes, respiratory and digestive systems. Osmium tetroxide is also a strong oxidising agent, and can cause undesirable damage or shrinkage of membrane components during the fixation process [14]. Sometimes paraformaldehyde in combination with glutaraldehyde is used due to faster penetration into the tissue, but this mixture exhibits a weaker fixing effect. 

### 2.2. Dehydration

Living organisms are largely composed of water, and it is necessary to dry them properly before placing them in a vacuum. Therefore, another important step in the preparation of biological samples for SEM analysis is dehydration, in which water present in cells is gradually removed. The dehydration process is carried out by immersing samples for several minutes in water solutions of increasing concentrations of ethanol or acetone, until reaching ethanol/acetone concentration of 100%. Ethanol is preferable as a dehydrating agent due to the fact that anhydrous acetone absorbs water more strongly from the atmosphere than from a specimen [9,14]. Additionally, it has been shown that ethanol dehydration causes only a minimal reduction in a cell size [21].

An interesting approach is the application of an organosilicon compound, hexamethyldisilazane (HMDS), as a drying agent exhibiting a reduced surface tension and an ability to crosslink proteins. HMDS treatment is supposed to improve a mechanical stability of a specimen, which is beneficial in the next steps of sample processing and imaging [22]. HMDS is typically used as a final dehydrating solution in the course of an ethanol/acetone dehydration process [23]. After the last HMDS wash, specimens are left in HMDS until the complete evaporation of the solution. Numerous studies confirmed that HMDS treatment prevents the occurrence of imaging artifacts, allows a specimen to preserve surface details, as well as preserves cell microstructure without shrinkage [24]. Therefore, this drying method can be successfully employed for the visualisation of delicate samples, for instance pollen grains [25], but also hepatic endothelial cells [22], porcine retina [23] and bacteria (*P. fluorescens*) [26]. The risk of changing surface properties of a specimen as a result of silanization [27], however, restricts the application of HMDS treatment for high resolution imaging. Another drawback of HMDS is its acute toxicity [28].

An alternative to conventional SEM imaging performed under reduced pressure is the use of an environmental scanning electron microscopy (ESEM), in which the pressure around the sample is increased to 10–20 torr. In this method, gas particles in the chamber are ionised facilitating a free flow of current, which in turn allows for imaging of wet and non-conductive samples. The obvious advantage of imaging biological materials with ESEM is the ability to collect an image with minimal sample preparation [29,30]. However, a negative effect of humidity is still observed at higher pressures, and the presence of condensed water layer on a sample can result in low contrast and poor visibility of fine details of cells. Moreover, applied conditions are at a burden for biological materials, therefore it is generally accepted that a single ESEM sample can be only viewed once [31]. According to these limitations, the major challenge associated with ESEM is to dry the samples while preventing structural damage to investigated materials. Common drying methods include critical point drying, freeze-drying and direct air-drying after dehydration with alcohol.

Critical point drying (CPD) takes advantage of the fact that at the critical point the liquid turns into gas, and this phenomenon is not accompanied by distorting forces. A previously used dehydrating agent is displaced with a liquid (carbon dioxide or freon) brought to a critical point at elevated temperature and pressure. Once the critical point is reached, the heat is kept at the critical temperature and steam is slowly released from the chamber until the vessel reaches atmospheric pressure [14]. CPD procedure allows to reduce surface charging and improve contrast of observed materials, however it is not devoid of drawbacks. CPD requires the use of an additional equipment, and some types of samples may show artifacts, cracks, and undergo significant shrinkage upon processing [8,32]. Although CPD was found to cause a 25–30% reduction in the diameter of dehydrated cells, high resolution SEM studies demonstrated that all structural components of cells retained their usual relationship [21].

In a freeze-drying method, a specimen in the aqueous phase is quickly frozen and transferred to a special chamber where the temperature is kept below −80 °C. The frozen substance sublimates to the gas phase and then is absorbed or removed by vacuum. The process of freeze-drying can take from several hours to several days depending on sample size, temperature, and pressure. The major advantage of this method is limited shrinkage, especially when compared with CPD. However, some disadvantages include the need for a special equipment, problems with rapid freezing and transfer of the sample, long time needed to dry biological samples, and the presence of a sediment remaining on the sample surfaces. Additionally, freeze-drying can cause distortions and damage due to the formation of ice crystals [8,32,33,34].

The above-mentioned drying techniques, although giving excellent results for certain biological samples, are not always suitable for examining every microorganism or tissue. Moreover, they require complex, specialised, and expensive equipment. Therefore, the easiest and most effective way for drying of biological samples is a simple air-drying [35,36]. Although air-drying carries the risk of an excessive shrinkage, cracking, and collapse of fragile structures such as cilia and flagella, the use of solvents such as ethanol or HMDS in the dehydration phase enables to apply air-drying without damaging tested materials [34]. 

### 2.3. Sputter-Coating

Being non-conductive materials, dehydrated biological samples usually cause charging problems in SEM. The charge accumulated on the sample disrupts the primary electron beam leading to image distortion and low contrast. Therefore, it is essential to cover the sample with a thin layer of a conducting material, thus increasing its surface conductivity [37]. Consequently, a dried sample is placed in a vacuum sputter-coater. The pressure in chamber is lowered by means of a vacuum pump, and an inert gas is introduced. Then, gas molecules are ionised and strike a charged heavy metal target. Thereby, some of the atoms are knocked out and can coat the sample. The preferred sputtering metals are gold, gold-palladium, platinum, iridium, and chromium, while the three latter are usually selected for a high resolution imaging. Iridium, particularly, is preferably chosen as a coating material for high magnification applications [38], mainly due to its stability, resistance to oxidation, and the ability to provide a virtually grain-less coating layer [39]. Apart from metals, also a carbon layer can be used to increase surface conductivity of samples, particularly those amenable for X-ray microanalysis [38]. This material, however, should be deposited by either ion-beam sputtering or vacuum evaporation, since in conventional direct current magnetron sputter coaters it tends to form non-conducting diamond-like carbon films.

### 2.4. Optimised Protocol for Sample Preparation

Obviously, different research groups have developed their own sample preparation protocols, suitable for the visualisation of particular biological samples with a resolution degree matching specific needs. The majority of current SEM protocols [40,41,42] involves two fixation steps, one with paraformaldehyde and/or glutaraldehyde, and the other one with osmium tetroxide. Washing of specimen is followed by ethanol or acetone dehydration, critical point drying and metal sputter-coating. When adopted, these protocols may allow for the visualisation of biological objects with a resolution of 5 nm. Undoubtedly, this level of detail is high enough to thoroughly investigate a cell’s ultrastructure. Still, the majority of morphometric descriptors, as described later in this review paper, do not require such a high resolution.

The complexity of aforementioned approach inspired researchers to investigate how the modification of a sample preparation protocol would influence the quality of imaging. For instance, Moran et al. [23] studied numerous sample preparation methodologies for imaging the ultrastructure of porcine retina, representing a delicate biological tissue. It was shown that the additional fixation with osmium tetroxide did not improve the quality of imaging, providing that samples were first fixed in a formalin solution. Additionally, a careful dehydration of specimens with ethanol and HMDS was found to provide similar results as CPD, with an additional benefit of no specialised equipment required, lower costs and time commitment [23]. In the light of the above, we decided to further investigate whether simplifications of a sample preparation protocol could allow for collecting SEM images with an acceptable image quality.

In our research, we used SEM imaging to visualise the morphology of two types of biological samples representing prokaryotic and eukaryotic cells: a model Gram-negative bacterial strain *Escherichia coli* (DSM 30083, U5/41), and a cultured cell model of central nervous system neurons, namely rat neuroblastoma cell line B-35 (ATCC^®^ CRL-2754™). The details of culturing both types of cells can be found in our previous reports [43,44,45]. According to the optimised sample preparation protocol [43,44,45], cells were fixed using 3% glutaraldehyde for 24 h, then washed three times with sterile distilled water. Subsequently, samples were dehydrated by immersing them for 10 min in the solutions of ethanol with increasing concentrations (30%, 50%, 70%, 80%, 90%, 95%, 99.8%), then dried for 24 h at 50 °C. To enhance the quality of imaging, dehydrated samples were sputter-coated with a gold layer to produce a 5 nm thick conducting film. The developed protocol is depicted in Figure 1. Although this processing method is not recommended for collecting high resolution SEM images (resolution < 5 nm), it allows for imaging the details of cell ultrastructure larger than 50 nm, which is enough to carry on a morphometric analysis as described further in this paper.

## 3. Morphometric Analysis of Model Prokaryotic Cells: *Escherichia coli*

As prokaryotic cells, bacteria represent a maximally simplified structure consisting of cell organelles and DNA loosely embedded in the cytoplasm, surrounded by a cytoplasmic membrane and a rigid cell wall [30,46]. It has been shown that although various species of bacteria may differ significantly in shape or size, the variation in cell dimensions (length, width, aspect ratio, volume, etc.) may carry useful information about their growth phase, growth rate, and nutritional conditions. Moreover, shape is a selectable feature that helps cells survive under various conditions. Bacteria react to the surrounding environment to adopt size and shape that are optimal for current environmental conditions.

SEM is widely used in microbiological analysis to assess the morphology of bacterial cells, their adhesion to the surface, as well as their tendency to form bacterial biofilms [43,44,45]. Moreover, SEM allows to estimate the number and distribution of microorganisms on the investigated surface [43]. The versatility of results that can be collected by SEM analysis makes this type of microscopy favourable for the evaluation of antimicrobial character of medical surfaces [43], effectiveness of new antibiotic agents [47] or bioactive materials [44,45]. What is more, a high resolving power of SEM allows obtaining reliable information on the state of microorganisms in their natural environment [47,48,49].

Cell length and cell width can be easily measured from SEM images by manually tracing the dimensions of individual cells (Figure 2A,B). Results are usually reported as the mean value of multiple measurements. Dividing bacterial width by length allows to determine aspect ratio, which is close to unity for circular cells and decreases when bacterial cells become elongated. The values of bacterial length (L) and width (W) can be used to calculate cell volume (V) using the following equation [50]:
(1)$$V=\frac{\pi W^{2}}{4\left(L-\frac{W}{3}\right)}$$

By looking at cell morphology, it is possible to distinguish critical elements of the control of the cell cycle and the viability of microorganisms [51]. For instance, *E. coli* mutants lacking in binding proteins are also known from their inability to produce enough septation proteins to accommodate their increasing diameter. Therefore, instead of dividing, they continue to grow in length and girth until they lyse [52]. The cell shape of many rod-shaped bacteria is determined by the cytoskeleton MreB protein, actin homolog, and ftsZ, tibulin homolog. It has been found that lowering the mreB level results in an increase in cell width, while decreasing the ftsZ level leads to an increase in cell length [53,54].

It has also been shown that many bacterial species exhibit a surface-to-volume (S/V) homeostasis. Bacteria tightly control their cell cycles to adjust their size and the proper time of division. The shape of most bacteria is constant, and their growth is associated mainly with a change in volume. For example, volume changes in *E. coli* cells occur at a different rate, but usually cells keep the same shape—a rod with a constant aspect ratio (approx. 0.25). Disturbances in a normal bacterial growth including genetic alterations, lack of nutrition, and the effects of pharmacological agents disrupt homeostasis, and often lead to a change in cell width or length, and thus also S/V. For instance, increasing the length of cells reduces the S/V ratio. Although the cells become thinner, they tend to maintain homeostasis by reducing their width [55,56].

## 4. Morphometric Analysis of Model Eukaryotic Cells: B35 Neuroblastoma Cells

Imaging of various types of animal cells is increasingly applied in both medical research and diagnostics, since the shape of a cell is closely related to its biological properties [57]. The morphological analysis of cells has many applications, including the elucidation of numerous physiological mechanisms. For instance, SEM imaging is used to study morphological changes during the cell cycle, to analyse the phenomenon of cell division, mobility and growth dynamics, as well as the apoptosis and necrosis [58,59]. Microscopic images of cells are extensively analysed in clinical applications, especially in oncology, due to the fact that the morphology of cancer cells differs significantly from that of healthy ones [60,61]. The use of a morphometric analysis makes it possible to correlate the shape of analysed cells with the progression of a disease, identify abnormalities enabling early detection of cancer cells, and predict the course of a disease. Therefore, the use of microscopic techniques is an effective method of assessing the pharmacological effects of many drugs, including anti-cancer agents. Much information can be obtained by observing the behaviour of cells under specific stress conditions. Shape analysis can also assist to identify certain pathologies in many tissues, as well as the transition of cells towards drug-resistant phenotype [62]. Analysing cell morphology also provides an opportunity to assess the progress in cell differentiation and correlate cell shape with its functionality [57]. As a consequence, SEM analysis is commonly employed in cytotoxicity and biocompatibility studies, enabling the characterisation of new biomaterials [61,63,64]. 

Morphometric analysis is particularly useful in neuroscience, since the analysis of neural cell morphology, branching or formation of complex cell networks is important in assessing the state and proper functioning of the brain [64,65]. Moreover, SEM imaging is also used to assess the ability to control regeneration, replacement, and stimulation of neural cells—features that are extremely important for the diagnostics and treatment of neurodegenerative diseases, such as Parkinson’s or Alzheimer’s disease. The degeneration and death of neural cells have enormous health consequences prompting a lot of innovative research in the field of neural engineering [66,67].

When designing new solutions for cellular applications, it should be borne in mind that cells are extremely susceptible to many environmental factors. For instance, external electrical stimulation with direct current has been shown to selectively increase the growth rate of neurites towards the anode [68]. Aside from electrical and biochemical signals, there are increasing numbers of studies pointing to equally important physical parameters of the extracellular environment during the development of the nervous system. These include physical forces, mechanical factors, substrate flexibility, nanotopography, scaffold geometry and stiffness. All of these signals act as physical and chemical cues for the formation and reconstruction of neural tissue, hence may significantly affect cell shape [69,70]. 

The topography of an extracellular microenvironment has been shown to influence cell morphology, but also provide guidance and affect cell differentiation. Therefore, a widely studied trend among biomaterials is the inclusion of topographic cues to control the behaviour of neural cells and support their regeneration. Until now, cells of the neuronal type have been grown on isotropic (microfilars) and anisotopic (meshes, microchannels, electrospun fibres) surfaces, as well as undefined random topographies that may better reflect natural conditions in tissues [71]. To relate topographic cues to specific cell responses, surface must be well analysed with respect to width, depth of the grooves and microgrids, the length or diameter of the fibres, and the shapes of the holes, etc. Moreover, not only the topography but also the dimensions of individual features play a significant role in the development of cells, since surface features which are similar in size to neurons can enhance cell–substrate interactions [66,70,72]. Consequently, SEM imaging of neural cells on the surface of biomaterials enables to predict and control cell adhesion, spreading, alignment, and morphological changes.

### 4.1. Conventional Morphometric Descriptors

A neuronal cell is characterised by the presence of elongated neurites, from which axons and dendrites can be distinguished as forming branched structures. The morphological complexity of the branches largely determines the functional capacity of cells. When neural cells are connected to each other, they form networks that serve as the basis of neural function. Shape development is determined both by genetic factors and interactions with a surrounding tissue. To fully describe the morphology of neural cells, a number of metric parameters are used, including soma size, neurite diameter, neurite length, cell area, cell volume, and correlations between them [73]. Another set of parameters is used to assess the development of neural network (branching descriptors), namely neurite density, neurite alignment, number of cells forming neurites, etc. [70,72,74,75].

Neurite number is defined as the number of neurites exiting a single cell body [76] (Figure 3A). In the case of branched cells, a typical descriptor is an average neurite length, defined as the sum of the lengths of all neurites in a single cell divided by the total number of neurites [75,77,78]. The measurement of neurite length, which affects networking and signal transmission, can be made by a manual tracing of the entire neurite from the base to its edge with the use of an image analysis software, for example ImageJ (NIH) (Figure 3B). When describing neural length, one can also use a maximum neurite length defined as the length of the longest neurite in each cell. By determining a radial distance (the minimum length from the cell to the ends of the neurite), it is also possible to assess the straightness of the branches by defining the difference between the length of the neurite and the radial distance [79].

Angular measurements are usually used to analyse the alignment of neurites on the surface of topographically modified material (Figure 3C). The purpose of such materials is to specifically target neural cells, thereby proposing a therapeutic solution for the regeneration of damaged cells. To estimate a directionality of the neurons, a straight line is drawn from one end of the neurite to the other one, and an angle is determined with respect to the grid axis. When the measured value of the angle is less than 15°, the neurite is considered parallel to the designated grids. On the other hand, when it is greater than 75°, the neurite is scored as perpendicular to the cues. In the case of substrates without a topographic pattern, a random direction can be selected as the axis [71,80].

SEM visualisation of neural cells is a preliminary indicator of interactions allowing the assessment of surface adhesion, which is crucial for cell survival and development. Subsequently, it is possible to assess the number of differentiated cells on tested biomaterials. This can be done by visually examining the field and counting cells that have at least one neurite equal to the diameter of a cell body [73]. The formation of neural networks, lengthening and branching of neurites is a reliable indicator of the functioning of neurons, therefore this analysis is widely used in the assessment of biological functionality of biomaterials. Moreover, it is also possible to evaluate the geometric complexity of the cells by expressing inter cell area, perimeter, or circularity [60,62,81]. An area of the cell is calculated as the area defined by the closed curve and presented by counting all pixels (Figure 3D). A perimeter is measured by the shape of the cell defined by the outline and by counting pixels (Figure 3E), and a circularity (C) is expressed as a function of cell area (A) and perimeter (P) with the use of the following formula [61,82]: (2)$$C=\frac{4\pi A}{P^{2}}$$

Circularity of 1 is typical for round cells, and it decreases for cells with irregular shapes. The analysis of these morphometric descriptors was found to be suitable to differentiate between different types of microglial cells present in the neocortex of an injured rat brain [81]. For instance, circularity was useful to discriminate bushy cells from all other cell groups, including ramified and hypertrophied microglia. Additionally, the analysis of circularity allowed for the detection of significant differences between hypertrophied and ramified cells from injured brains. Interestingly, changes in cell perimeter and cell area were found to be more informative than circularity. Numerous studies report on the changes of a microglia perimeter as a result of phagocytosis [83] or pathological situations, including traumatic brain injury or inflammatory reaction induced by neuraminidase [82].

### 4.2. Fractal Analysis

Apart from conventional descriptors of cell morphology, fractal analysis has recently become a recognised technique to characterise neural tissue [84,85]. Introduced by Mandelbrot in 1977 [86], fractals are characterised by a scale-invariant and self-similar behaviour. This type of geometry is abundant in nature, with numerous examples in anatomy, including cardiovascular, respiratory, and neural systems [84]. Since fractal connectivity is known to be a basis for brain organisation and complexity, fractal analysis allows to quantify complex patterns found in neuroscience and to make predictions about clinical outcomes [85]. The most frequently used fractal descriptors are fractal dimension, *D* (defined as a statistical index of complexity comparing how a detail in a cell changes with the scale at which it is measured) and lacunarity, *L* (defined as a measure of how cells fill the available space).

Fractal analysis can be performed with the use of different types of images, including SEM micrographs [84], confocal micrographs [84], fluorescence images [87], magnetic resonance images [88], etc. The starting point for a typical fractal analysis consists of thresholding the image to eliminate background noise and is followed by converting the image to a binary format and outlining the shape of a cellular network. As-formed binary silhouettes of the neurons, as depicted in Figure 4, are further quantified with the use of different algorithms. Depending on the magnification, fractal analysis could be performed either for a single neuron or for a neural network. Even though fractal analysis of a neural network could be done using optical imaging, this method would not be useful to visualise features on optically nontransparent materials. The use of SEM allows to perform fractal analysis of cells on a variety of substrates, regardless of their degree of transparency [34].

Fractal analysis is particularly suitable for the investigation of dendritic arborisation, which is defined as a process in which neurons create new synapses by forming new dendritic trees and branches [89]. In this context, a small value of D should be correlated with cells exhibiting a simple, uncomplicated structure, for example those in the early stage of development [90]. On the other hand, cells exhibiting complex structures with a high level of branching exhibit a large value of D. In a recent work, Smith et al. [84] analysed three-dimensional SEM images of rat neurons to investigate the degree to which neurons resemble fractals, as well as the origins of this fractality and its impact on neuron functionality. The results showed that different types of neuron were characterised by different values of D, and, in general, D was higher for neurons with a greater need for connectivity, for example hyperbranched Purkinje cells [91]. D was also found to be affected by pathological states of neural tissue, for example those associated with Alzheimer’s disease [92]. The disturbances in fractal branching decreased the degree of interconnections between neighbouring neurons and limited transfer of nutrients and energy.

Although D is suitable for detecting the changes in branching structure, it is not sensitive to some features that are hard to visually recognise, e.g., the relation of soma size to process length [93]. In this case, L can be used to distinguish similarly looking microglial morphologies with the same D. In fact, the term lacunarity was introduced to describe the gaps between various features of fractal objects, allowing to distinguish between different texture details [94]. Therefore, L is suitable to assess the degree of clustering within a neural network, as well as to distinguish between dense structures and scattered/non-connected objects [88]. Different values of L calculated basing on SEM images of two neural cell populations may indicate that either these cells originate from different parts of the brain, or one group represents damaged cells. For instance, L was found to be an indicator of Alzheimer’s disease, as this pathology was manifested through the presence of lacunar formations in the brain [95]. Additionally, L was found to increase with the development of a disease, resulting in an anomalous functioning of the brain.

### 4.3. Textural Descriptors

Recent advances in digital imaging and computing allowed for the development of imaging processing tools aimed to quantify the perceive texture of an image. In a general meaning, image texture provides information about spatial arrangement of intensities (grey bands) in an image [96] and can be qualitatively described with the use of following descriptors: regularity, directionality, fineness, coarseness, smoothness, granulation, randomness, lineation, being mottled, irregular, or hummocky [97,98]. The use of these descriptors is usually called “perceptual characterisation” and is useful for a coarse classification of textures [98]. A quantitative description of a texture requires the use of advanced computational methods, for instance a multiresolution decomposition using Gabor wavelets [98], artificial neural networks [99], as well as a textural analysis based on a gray-level co-occurrence matrix (GLCM) [100].

Since SEM enables to capture fine shapes of biological objects, this technique has been routinely used to collect images suitable for the analysis of textural descriptors [99,100,101,102]. Still, it should be borne in mind that the processing method could affect the texture of a sample. For instance, too harsh dehydration could result in the formation of defects in the plasma membrane of mammalian cells. The presence of cracks and crinkles has been previously noted for samples after freeze-drying, particularly those not treated with a fixing agent [34]. Actually, using glutaraldehyde and ethanol dehydration strengthens the surface of a cell through cross-linking the collagen present in the extracellular matrix.

To perform a quantitative analysis, however, a high number of SEM micrographs should be collected and processed point by point [99]. The most popular texture analysis method, GLCM, allows to present surface details of a sample with a mathematical function (procedural texture), and to extract the following texture descriptors: angular second moment, entropy, correlation, contrast, and inverse different moment [99,103]. A set of texture descriptors can be then applied as a base for a high efficiency classification tool that will allow to recognise elements of an SEM image on the basis of the texture of image pixel.

So far, there are only few literature studies reporting the application of textural analysis in biological sciences, including the detection of polysaccharide in raspberry powders [99], and cervical precancerous [102]. Nevertheless, the presence of different textures in SEM images of B35 cells, associated with a cell’s body, an extracellular matrix, neurites, etc. (Figure 5), suggests that the textural analysis should be also suitable for the analysis of neural cells.

## 5. Conclusions

Scanning electron microscopy appears as an extremely useful tool to analyse morphology of different types of cells, including prokaryotic and eukaryotic ones. Providing that a biological sample is prepared with special care, for instance using the optimised protocol presented in this study, SEM allows to achieve the resolution of 50 nm, which is more than enough to investigate cell morphology in detail. Therefore, SEM can be efficiently used to monitor even slight variations in cell dimensions and morphology, which could be related with the onset of important changes in the state of cells. Due to the versatility of morphometric descriptors and their relation with molecular processes, it is possible to apply SEM as a convenient screening method to detect cell pathology.

## Figures and Tables

**Figure 1 cells-10-03304-f001:**
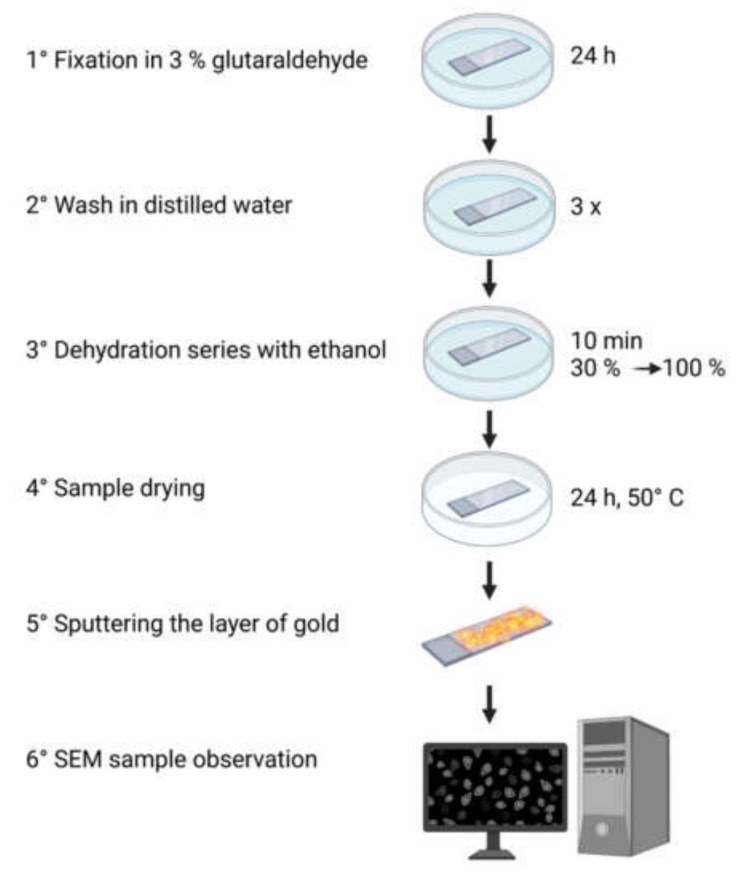
Schematic representation of the optimised protocol for the preparation of *Escherichia coli* and B-35 cells for SEM imaging.

**Figure 2 cells-10-03304-f002:**
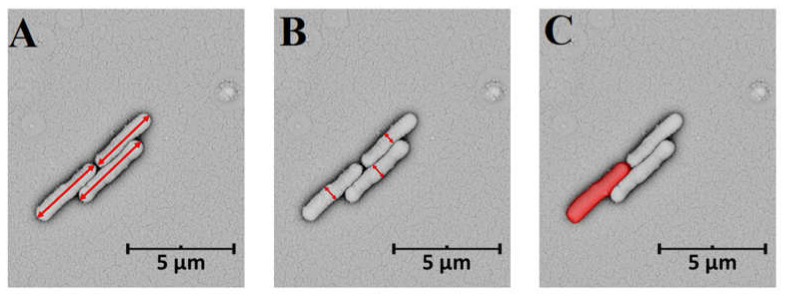
Morphometric descriptors of bacteria on the example of *Escherichia coli* cells: (**A**) length, (**B**) width, (**C**) cell surface area. SEM micrographs were collected by means of a Phenom ProX scanning electron microscope operating at 15 kV.

**Figure 3 cells-10-03304-f003:**
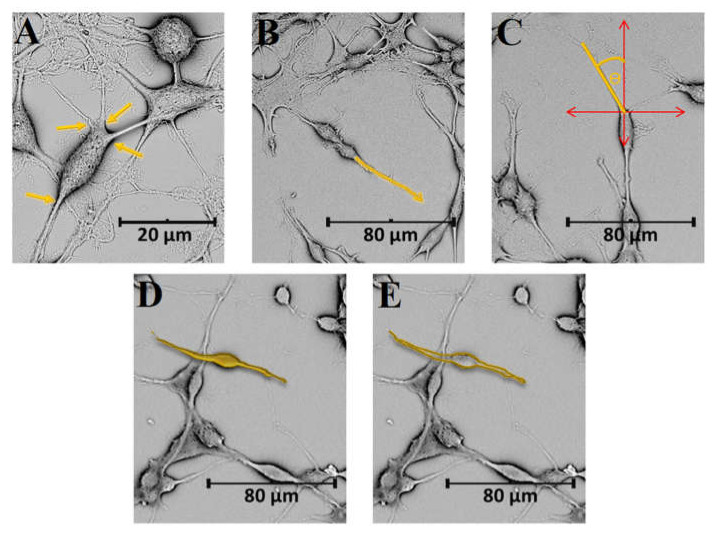
Morphometric descriptors for neural cells on the example of rat neuroblastoma B35 cells: (**A**) number of neuritis, (**B**) length of neuritis, (**C**) angle, (**D**) cell area, (**E**) perimeter. SEM micrographs were collected by means of a Phenom ProX scanning electron microscope operating at 10 kV.

**Figure 4 cells-10-03304-f004:**
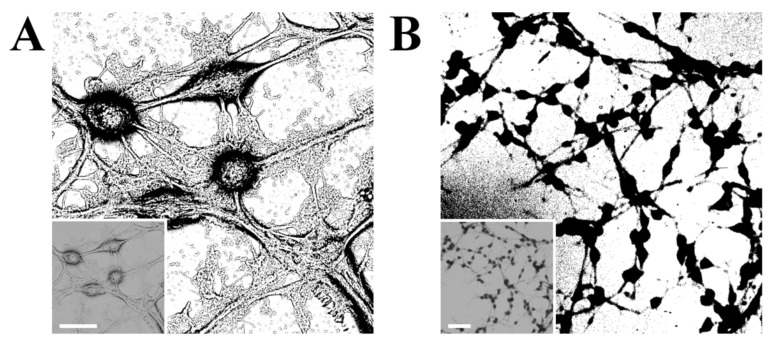
Binary images of B35 cells in (**A**) high magnification (scale bar is 20 µm), and (**B**) low magnification (scale bar is 100 µm), with source SEM micrographs (Phenom ProX, 10 kV) as the insets.

**Figure 5 cells-10-03304-f005:**
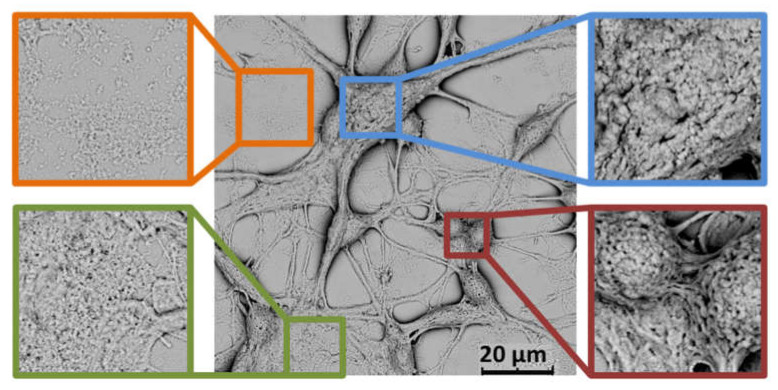
SEM images showing different textures typical for neural cells on the example of rat neuroblastoma B35 cells. The defects present on the surface of cells should be related to a dehydration process. SEM micrographs were collected by means of a Phenom ProX scanning electron microscope operating at 10 kV.

**Table 1 cells-10-03304-t001:** Comparison of the most popular fixatives used for processing of biological materials for SEM imaging; where dH_2_O is deionised water, CAC is a cacodylate buffer; PB is a phosphate buffer.

Fixing Agent	Fixing Conditions	Fixing Time	Rinsing Conditions	Major Risks	Ref.
Osmium tetroxide	1% OsO4 in dH2O or 0.1 M CAC	30–60 min	1 × 2 min with 0.1 M CAC or dH2O, and 2 × 2 min with dH2O	Causes eye and skin burns. Causes digestive and respiratory tract burns. Aspiration hazard if swallowed. Can enter lungs and cause damage. May cause adverse reproductive effects. **Target organs**: kidneys.	[9,15,16,17]
	0.5% OsO4 and 0.8% K4Fe(CN)6 in dH2O or 0.1M CAC (reduced osmium)	30–60 min	1 × 2 min with 0.1 M CAC or dH2O, and 2 × 2 min with dH2O	Causes eye and skin burns. Causes digestive and respiratory tract burns. Aspiration hazard if swallowed. Can enter lungs and cause damage. May cause adverse reproductive effects. **Target organs**: kidneys.	[9,15,16,17]
Glutaraldehyde	1.5–4% in 0.1 M CAC or PB, pH 6.8–7.4	20–60 min for animal cells, 1–48 h for bacterial cells	3 × 2 min with 0.1–0.2 M CAC or PB	Causes eye and skin burns. Causes digestive and respiratory tract burns. May cause allergic respiratory and skin reaction. Harmful if swallowed, inhaled, or absorbed through the skin. Aspiration hazard if swallowed. Can enter lungs and cause damage. Dangerous for the environment. **Target organs**: central nervous system, lungs, respiratory system, eyes, skin.	[9,14,15,16,18]
Paraformaldehyde	4% in 0.1 M CAC or PB, pH 6.8–7.4	30–60 min for animal cells, 48 h for bacterial cells	4 × 5 min with 0.1 M CAC or PB	Harmful if swallowed. Causes skin irritation. May cause an allergic skin reaction. Causes serious eye damage. Harmful if inhaled. May cause respiratory irritation. Suspected of causing cancer. **Target organs**: respiratory system.	[9,16,19]
Methacarn	methanol/chloroform/acetic acid 6:3:1	48 h for bacterial cells	4 × 5 min with 0.1 M CAC	May cause irritation to the eyes, nose, throat, headache, dizziness, nausea. **Targe****t organs**: eyes, skin, respiratory system, central nervous system, gastrointestinal tract.	[16,20]

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
