# Peer review of "Guidelines for a Morphometric Analysis of Prokaryotic and Eukaryotic Cells by Scanning Electron Microscopy"

_cells, 2021, doi:10.3390/cells10123304_

Round 1

Reviewer 1 Report

The paper is well written and clearly describes the sequential steps necessary to optimize sample preparation for high-resolution  SEM mages.  

Author Response

We would like to thank the Reviewer for a positive opinion on our work.

Reviewer 2 Report

In this manuscript, the authors present an interesting historical overview of scanning EM and its contribution to biological science. They also describe an optimized protocol for the fast and simple preparation of biological specimens of two different classes, and provide examples for morphometric analysis of prokaryotic and eukaryotic cells. Overall, this work is worthy of publication and could contribute to further integration of electron microscopy in the biological sciences. However, several issues need to be addressed before it can be accepted for publication in Cells.  

The manuscript is divided into three major sections: “Sample preparation”, “Morphometric analysis of model prokaryotic cells: Escherichia coli”, and “Morphometric analysis of model eukaryotic cells: B35 neuroblastoma cells”. While the sections on morphometric analysis present extensive background and offer useful examples for measurements and calculations of various parameters, the section on sample preparation seems to have been given less thought. This section includes an over-simplified protocol, with minimal background, that may not be adequate for most biological applications. The authors should honestly present the limitations and weaknesses of this approach and may even do better by providing an optional addition of an extended sample preparation path.

Major concerns:

  1. The authors discuss the necessity of careful sample preparation (line 13) and claim that their protocol “allows to collect high-quality images with a minimum processing effort” (line 14), but this is an inherent contradiction. The terms “high quality” or “high resolution” should be more objectively defined (for instance, regarding a magnification range or observable details of ultrastructure). In practice, it may only be possible to obtain images that are adequate for a specific type of analysis. See also points 2-4, below.
  2. The authors rightly state that: “Inadequate preparation of specimen may result in shrinkage and deformation of cells, leading to incorrect results and misleading interpretations” (line 62). Despite this, they advocate the use of a protocol that bypasses critical point drying (CPD). The majority of current SEM protocols use the CPD technique in order to obtain “real” high quality images; see for example: Goldberg & Allen, J Cell Biol. (1992), Bray, Supercritical Fluid Methods and Protocols (2000), Hover et al., Appl. Environ. Microbiol. (2016). Again, the authors should more openly discuss the pros & cons of taking the shortened approach.
  3. Gold, gold-palladium, platinum and chromium are mentioned as preferred sputtering materials (line 150) with the latter two singled out for “high resolution imaging”. In fact, iridium is a superior coating metal for this purpose, albeit with the requirement for prior CPD treatment. See for example: Fichtman et al., Methods Cell Biol. (2014) and subsequent publications citing this protocol.
  4. The authors claim that their protocol achieves a “resolution of several nm” (line 359), however: sputter-coating of the sample with a gold layer for 20 min at 20 mA will likely produce a granulated layer of gold that is dozens of nanometers thick! This will considerably affect the accuracy of measurements and the quality of images.

Minor concerns:

  1. The authors propose that using haxamethyldisilazane in the dehydration phase allows “air-drying without damaging tested materials” (line 140), however haxamethyldisilazane changes the surface properties of the sample (by silanization) and is not advisable for a high-resolution imaging.
  2. Osmium tetroxide is mentioned as harmful for health, but aldehydes and hexamethyldisilazane are also very harmful materials and this should be stated clearly.
  3. The written text is quite clear and easy to follow, but it’s full of grammatical errors in English. The most common kind of these errors is a missing definite article, as in: “…proper preparation of (the) sample is essential” (line 60) and there are many errors concerning indefinite articles (aan). The whole manuscript should undergo professional proofreading.
  4. In Line 247, please change “neuritis” (inflammation) to “neurites” (plural for neurite – neuronal processes).

Author Response

Major concerns

Answer 1:

We agree with the Reviewer that our claim was not precise. Therefore, we have corrected the abstract and main text (page 2) to provide a minimum size of objects that can be observed if the sample is prepared according to the simplified protocol (50 nm).

Answer 2:

We agree with the Reviewer that CPD is required to obtain high quality images, as clearly presented in the literature studies provided by the Reviewer. Therefore, we have added this information to the section describing an optimized protocol for sample preparation (page 5-6). According to recent studies [1], however, it is possible to modify a general sample preparation protocol, while maintaining high quality of imaging. We have described these approached in the manuscript (page 5-6). Encouraged by the Reviewer, we have also included the limitations of the shortened approach of sample preparation (page 5-6), to make the Readers aware of the advantages and disadvantages of a sample preparation method described by us.

Answer 3:

According to the suggestion of the Reviewer, a paragraph describing sputter-coating of samples has been extended with a description of iridium and its properties that make it as a metal of choice for high magnification applications.

Answer 4:

We would like to thank the Reviewer for highlighting this issue. Actually, the achievable resolution is 50 nm, so we have revised the manuscript and corrected this information. According to our measurements, sputter-coating of a sample with a gold layer for 20 min at 20 mA results in the formation of a 5 nm thick Au layer. Although a granulated layer is formed, the sizes of granules are 10 times lower than the achievable resolution, so they are not expected to affect the accuracy of measurements.

The information about a thickness of a gold layer has been added to the manuscript (page 6).

Minor concerns

Answer 1:

According to the suggestion of the Reviewer, the limitation of using HMDS has been added to the section describing this drying agent (page 4).

Answer 2:

We agree with the Reviewer that safety is an extremely important issue, therefore, we have devoted a separate table (Table 1)  to compare popular fixatives used for processing of biological materials for SEM imaging (osmium tetroxide, glutaraldehyde, paraformaldehyde, as well as a mixture of methanol/chloroform and acetic acid), highlighting major risks and hazards associated with these chemicals. Besides, toxicity of HMDS has been also clearly stated in the section describing this drying agent (page 4).

Answer 3:

The manuscript has been proofread, and improved, according to the suggestion of the Reviewer.

Answer 4:

The change has been made as suggested.

Reviewer 3 Report

CzerwiÅ„ska-GÅ‚ówka and Krukiewicz summarize the use of SEM for morphological analysis of biological specimen, which is increasingly used for inferring the physiological states of the sample (prior to sample preparation). They start the review by cataloging current protocols in SEM sample preparation with an emphasis on best preserving the native structures of the sample by minimizing distortions at each step of the process. They then describe approaches to determining morphological descriptors, including both conventional descriptors and more recently used fractal analysis, and how these descriptors may report the underlying biology. Overall, the review is written in clear and plan language and is reasonably easy to follow. The content covered in this review is also relevant, concise, and potentially helpful to broad readers. I only have one major issue and a few minor ones, which I hope that authors will be able to address in the revision.

Starting with the major issue - the authors may wish to expand the fractal analysis sections by defining basic terms including fractal (especially what it means in the context of neurite morphological analysis), fractal dimension, as well as lacunarity. They also need to explain how the analysis is typically done and (at least speculate) how are fractal dimensions and lacunarity connect to neuron biology. Lastly, are these done based on SEM images? Is it possible to include some examples as in Figure 3 where conventional descriptors are illustrated?

There are also places where there is a gap in the text flow. For example, between lines 34 and 35. Right after describing the light source the authors went on to say ‘This allows for the formation of various types of signals …’. It is not clear at all how the use of electrons would naturally lead to the different types of signals; apparently some descriptions of how electrons interact with biological materials would be necessary to fill in the gap.

In the introduction (lines 50-51), they stated that ‘… a protocol that allows to collect best quality images of various biological samples …’. I am not sure if this is appropriate, as it is more likely that different biological samples require different preparation protocols for SEM. It is also not necessary to make statements like this.

Line 66, both carbon coating and metal coating can be used to increase sample conductivity and reduce charging.

Author Response

Answer 1:

The section describing fractal analysis has been expanded according to the suggestion of the Reviewer. Fractal dimension and lacunarity have been defined in the context of neural cells, and connected to neuron biology. The protocol of fractal analysis has been briefly introduced, and a figure has been added (Figure 4) to illustrate processing of SEM images (binarization). The list of imaging techniques that can be used to perform fractal analysis has been provided (the list includes SEM).

Answer 2:

We have revised the aforementioned sentence and we hope that now the information is clear.

Answer 3:

We would like to thank the Reviewer for this comment. According to the Reviewer’s suggestion, we have modified the aforementioned sentence (page 2).

Answer 4:

The information about using carbon for coating the specimens has been added to the relevant section (page 5).

Reviewer 4 Report

The manuscript by CzerwiÅ„ska-GÅ‚ówka and Krukiewicz entitled “Guidelines for morphometric analysis of prokaryotic and eukaryotic cells by scanning electron microscopy” describes the use of 2D SEM images of biospecimens for morphometric analysis, a powerful indicator of cellular function.

Major comments:

1) Unfortunately, this manuscript does not describe any novel information in the field and the authors fail to discuss reliability of measurements and interpretation of the resulting analysis based on specimen preparation utilized.  The authors also fail to analyze any textural descriptor which has been shown to increase the accuracy of biospecimens measurements, especially considering the use of SEM in this study. 

2) The imaging conditions, equipment utilized, thickness and material used for the specimen’s coating should be included.

3) The dehydration protocol seems extremely harsh on the sample and the authors should include measurements of the specimens before and after this step to evaluate any defects introduced by the sample preparation method. Percentage of shrinkage is the major concern as well as anisotropic deformations of the mammalian cells.

Author Response

Answer 1:

We would like to thank the Reviewer for an extensive evaluation of our work. In the revised version of the manuscript, we have expanded the section devoted to sample preparation (page 4-6), indicating the limitations of a described approach and presenting alternative processing techniques. A discussion of how the modification of a sample preparation protocol affects the quality of imaging has been also added (page 5-6). We have also included the analysis of textural descriptors, as suggested by the Reviewer (page 11-12).

Answer 2:

We agree with the Reviewer that the detailed description of an experimental design is a fundamental part of any research paper. In our review paper, however, SEM images are used as visual representations of morphometric descriptors used to analyse either bacteria or animal cells. Therefore, instead of providing a materials and methods section, we have added the literature references containing all details of cell culture studies, sample preparation, as well as imaging. The information about thickness and material used for specimen’s coating (a 5 nm Au layer) has been added to section 2.4 Optimized protocol for sample preparation (page 5-6), and the details of imaging conditions and equipment have been added to figure captions.

Answer 3:

The issue of cell and organelle shrinkage during preparation for scanning electron microscopy has already been studied in the literature [1–3]. Gusnard et al. [1], for example, compared several fixation and dehydration methods of mouse hepatocyte nuclei and human erythrocytes. Although critical point drying caused most of the observed shrinkage (a 25-30% reduction in diameter in both specimens), high resolution SEM studies demonstrated that all structural components of cells retained their usual relationship. Interestingly, both glutaraldehyde fixation and ethanol dehydration caused only minimal size reduction. The significantly remarkable preservation of cell microstructure without shrinkage was obtained by HMDS [2].

This information has been added to the relevant sections of the manuscript devoted to sample preparation (page 3-5).

References

[1]      D. Gusnard, R.H. Kirschner, Cell and organelle shrinkage during preparation for scanning electron microscopy: effects of fixation, dehydration and critical point drying, J. Microsc. 110 (1977) 51–57.

[2]      S. Nikara, E. Ahmadi, A.A. Nia, Effects of different preparation techniques on the microstructural features of biological materials for scanning electron microscopy, J. Agric. Food Res. 2 (2020) 100036.

[3]      Y. Zhang, T. Huang, D.M. Jorgens, A. Nickerson, L.-J. Lin, J. Pelz, J.W. Gray, C.S. López, X. Nan, Quantitating morphological changes in biological samples during scanning electron microscopy sample preparation with correlative super-resolution microscopy, PLoS One. 12 (2017) e0176839.

Round 2

Reviewer 4 Report

Minor comments

- Page 4 line 102 “organisilicon” should be corrected to “organosilicon”

- Page 4 line 115 “HDMI” should be corrected to “HDMS”

- Page 5 line 171, iridium has been used for decades in HRSEM imaging

Major comments:

- The “holes” observed on the plasma membrane of mammalian cells under study (Figs 4-5) are indicative of dehydration defects. These defects will affect the textural analysis of the images, the authors should comment on this concern.

-In Figure 4 the magnification utilized to acquire the SEM images is low and comparable to an optical microscope type of image, therefore it is not clear the advantage of utilizing this method for neural network analysis.  Can the authors achieve similar results avoiding the entire sample preparation method by using optical imaging?

Author Response

We would like to thank the Reviewer for the extensive evaluation of the revised version of our manuscript, and for picking some minor and major issues that had to be corrected.

We have followed the suggestions of the Reviewer, and we have corrected spelling mistakes (organisilicon, HDMI), as well as we have clarified the role of iridium in HRSEM imaging (page 5). Also, we have extended the discussion on the role of dehydration on receiving high quality images suitable for textural analysis (page 11-12). Last but not least, we have corrected the section describing fractal analysis of SEM images: Fig.4 was redrawn to visualise the ability to use SEM for fractal analysis of either single cells or neural networks, as well as a major advantage of SEM over optical imaging, namely possibility of imaging optically nontransparent materials, was mentioned (page 10).